# The View of the French Dog Breeders in Relation to Female Reproduction, Maternal Care and Stress during the Peripartum Period

**DOI:** 10.3390/ani10010159

**Published:** 2020-01-17

**Authors:** Natalia R. dos Santos, Alexandra Beck, Alain Fontbonne

**Affiliations:** 1Unité de Médecine de l’Elevage et du Sport (UMES), Ecole Nationale Vétérinaire d’Alfort, 94700 Maisons-Alfort, Paris, France; alain.fontbonne@vet-alfort.fr; 2Ceva Santé Animale, 33500 Libourne, France; alexandra.beck@ceva.com

**Keywords:** survey, maternal care, behavior, peripartum, stress, breeding, purebred

## Abstract

**Simple Summary:**

Recent research has shown that the quality of maternal care in dogs has both direct and indirect effects on the health, welfare and social development of puppies. The importance of these findings, however, is not yet being fully acknowledged by dog breeders. To evaluate French dog breeders’ current understanding and management practices as related to maternal behavior and stress during the peripartum period (time just prior to, during and after birth), an online survey was developed. Four major components of kennel activity were surveyed, including breeder demographics, pre- and post-birth management of females, and changes in maternal behavior that were observed during this time period. Overall, the survey indicated that, in France, dog breeding enterprises are usually family-based and contact between animals and breeders is fairly common. Breeders did feel that the peripartum period was stressful for females. The most common technique used to reduce anxiety during this critical time was to reassure the bitches by increasing human presence and contact.

**Abstract:**

In France, as in many other western countries, dogs are an important part of the society as pets or working animals. The exact demand for puppies in France is unknown, as is the proportion of dogs coming from different breeding sources. Nevertheless, the origin of puppies is important since young dogs from puppy mills and pet stores appear to be more likely to develop behavioral disorders. Puppies coming from responsible breeders, on the other hand, tend to be better adapted. In well-managed kennels, an explanation for these behavioral differences may be associated with greater contact of litters with the dam and humans. Another factor that might influence a dog’s temperament and character is maternal behavior. Although recent studies have shown that the quality of maternal care in dogs is important, direct effects on the development of behavioral problems such as fearfulness or noise sensitivity are still controversial. To better understand the view of breeders, due to an increased interest in maternal care of dogs, an online questionnaire was developed to assess the dog breeders’ profiles and to explore their perception of normal maternal and stress-related behaviors during the peripartum period. In addition, the management of the female during this critical time was queried. Three-hundred and forty-five French dog breeders, representing 91 breeds, completed the online survey. Considering the demographics of the responders, breeding activity in France is mostly family-based with 76% raising two breeds of dogs that produce, on average, five litters/year. Around 60% of the breeders use progesterone levels to determine breeding date. The whelping date is estimated using multiple criteria and most labors and deliveries happen under human supervision. The majority of behaviors associated to good maternal care are defined by the vast majority as being related to more attention of the bitch towards the puppies with the frequency of nursing and licking being important clues. Globally, the peripartum is perceived as a stressful period and to minimize stress and reassure the bitch the favored method used is increasing human presence. Problems related to maternal behavior were described, notably with primiparous bitches.

## 1. Introduction

Dogs, either as companions or working animals, are an important part of global society. In France, dogs also play a significant role in public life. With 6.95 million dogs residing in the country in 2018 and a high ratio of dogs to humans (17/100), France ranks fifth in the European Union in terms of number of dogs owned [1]. To better track dog numbers, a law has been enforced since 1999 where all dogs must be identified by a microchip or tattoo. In 2013, a national database for identification of domestic carnivores (*Identification des Carnivores Domestiques*—*ICAD*) was implemented. This database manages all the identification documents of dogs and provides a system for tracing identification numbers. In addition, the French Kennel Club (*Société Centrale Canine*—*SCC*) is responsible for registering purebred puppies born in France. However, there are considerable differences between the number of puppies identified by ICAD and those registered by SCC. In 2016, ICAD identified 749,720 new dogs, with puppies comprising over 90%, compared to 227,993 puppies registered by the SCC. This discrepancy of nearly half a million puppies is explained by the fact that puppies recorded by ICAD are not purebred and may or may not have been born in France. Although the demand for new dogs or puppies in France is not known, database numbers indicate that the production of puppies is an important pillar of the dog breeding industry.

The origin of puppies is an important consideration because puppy source has been suspected to be associated with behavioral issues. A summary of published data on behavioral problems shows that dogs from puppy mills and pet stores tend to have an increased incidence of behavioral disorders such as fear and aggression towards other dogs and humans [2]. Moreover, behavioral disorders are also a major reason for euthanasia, abandonment [3] and sheltering of pet dogs [4]. Although it is challenging to identify each factor’s contribution to the behavioral development of dogs, this subject has been an important area of research. While it is hard to characterize all the elements that contribute to making a well-balanced and stable dog, a better understanding of the breeders’ operational features and vision in relation to maternal behavior, to the impact of stress levels during the peripartum period and stress management could potentially improve the wellbeing of dogs. Over the years, some conditions have been studied and deemed to be essential including the socialization of puppies during early life [5,6,7,8,9], genetics [10] and responsible ownership [11]. Optimal environmental surroundings adapted to the individual dog’s level of development [12], the duration of maternal-puppy interaction [2] and puppies age at weaning time [13,14] also may influence the dog’s temperament. Due to the significance of the source of the puppies, it is important to better know the breeding environment as well as the breeders’ view on the importance of reproduction and maternal behavior to optimize the production of well-balanced dogs. Even though the opinions and operational practices of dog breeders across the world are very important, these factors have received little attention [15]. The selection of breeding stock and the breed goals of purebred dogs have been documented in France [16] and Australia [15]. To our knowledge, no study has ever investigated current breeding operations and strategies, although a single study [15] has addressed the importance of selecting females based on maternal behavior. Because of increased attention to maternal care, we developed a questionnaire to assess the operations of dog breeders in France and to explore breeder perception of normal maternal behavior, stress-related behaviors and stress management during the peripartum period, including whelping.

## 2. Methods

### 2.1. Web-Based Survey

Recruitment of breeders was done via email using lists that included clients of the Centre d’Etudes en Reproduction des Carnivores (CERCA-EnVA) (*n* = 107), participants of a dog roadshow co-organized in 2017 by Ceva and Royal Canin (*n* = 1140) and Dogs Revelation magazine social media (*n* = 10,585 followers). The survey was accessed via an internet link (see Appendix A). The questionnaire was self-explanatory with a brief introduction that described the purpose of the study. 

The survey, available from March 2018 until July 2018, consisted of five sections. The first recorded the contact details of the breeder with emphasis on the type of activity (professional or family), the numbers of breeding bitches and litters per year, and the dog types and specific breeds represented in each kennel. In addition, information regarding animal housing systems and the level of interaction with humans was collected. The second section concerned the breeding program. Questions focused on the use of specific techniques to estimate breeding and whelping times and changes in housing, exercise and other alterations of the bitch’s routine around parturition. The third section was related to whelping, with requested information about the overall time of parturition, if whelping was monitored, whether signs of maternal stress were observed and what, if anything, was done to address peripartum stress. The fourth section contained information related to maternal behavior. Queries were made on how much time the female needed to adapt to her maternal role, signs of poor maternal care, methods used to reduce the stress and improve maternal care, common behaviors observed around parturition and the frequency of problems with maternal behavior. Section four also included responses to questions regarding observations made for specific maternal behaviors, such as growling towards the owner or strangers, cannibalism, rejection of puppies and insufficient lactation. In addition, questions were included on the number of weeks the puppies usually remained with their mother and what qualities were perceived to be indicative of good maternal instincts. The fifth and final section was open-ended. That is, breeders could express their thoughts about maternal behavior in general and provide comments on any observed behavioral differences among breeds. Overall, this survey consisted of 35 questions that could be completed in approximately 15 min. Responses to all questions were required to submit the survey (no blank responses were allowed). Finally, to ensure that only breeders responded to the questionnaire, the identity of a random sample of thirty percent of the participants was verified based on the information available from breeding associations, and kennel status was confirmed by checking for an active professional identification number.

### 2.2. Statistical Analysis

Descriptive statistics were used to characterize relevant parameters. Categorical variables were expressed as frequency counts and percentages. Cross analyses were performed to further describe the categorization of the activity, breeding population size, number of litters per year, the accommodations provided inside houses and/or kennels, and the methods used, both to monitor and reduce stress at whelping. All categorical variables were analyzed using the Chi square test or the Fisher exact test. All tests were performed with type one error (α) set at 5% and the SAS software package (version 9.3) [17] was used for all statistical analyses.

### 2.3. Ethics Statement

In accordance with the General Data Protection Regulation (GDPR), the respondents were free to choose whether to participate in the survey and the data obtained only addressed the stated objectives of the research. A statement was provided in the survey introduction to assure that responses were confidential and the information collected would be used for research purposes only. In accordance with the regulations on personal data, any respondent has rights of access and rectification and limitation of processing of their data. They may also oppose the processing, withdraw their consent and request the deletion and portability of their data by completing a form available from Ceva (https://www.ceva.com/en/Footer-s-links/Privacy-policy3#contact).

## 3. Results

### 3.1. Breeder’s Profile and the Kennel Organization

Three-hundred and forty-five French dog breeders completed the survey and none of the questionnaires were discarded due to being incomplete. 

With regard to activities, 58.8% (203/345) classified their breeding enterprises as familial, with 28.4% (98/345) reporting that breeding was their main professional occupation. The majority of breeders (93.9%; 324/345) raise up to three different breeds with 76.5% (264/345) raising a single breed. In relation to the breeding stock, 84.3% (291/345) of the responders have up to 10 breeding bitches and 74.5% (257/345) have less than five litters per year. Further data have been analyzed according to the size of the facilities, using 10 as a threshold, based on French regulatory requirements for “Classified Installations for Environmental Protection”. 

All breed groups were represented with a larger number of herd dogs, followed by pet breeds (Figure 1). 

### 3.2. Housing System

Dogs were most commonly housed in close proximity to the family with a high level of human contact (Figure 2 and Figure 3), with most bitches living inside or having access to the house during some parts of the day (66.4%). The remaining third of the respondents kept their animals in a kennel and/or a garden. The type of housing system employed was correlated to breeding activities (*p* < 0.0001): when bitches were housed exclusively at home (*n* = 150), breeders that considered their operation to be a family activity were over-represented (78%), compared to professional breeding kennels (10.6%). On the other hand, breeders lodging bitches in a garden or in a kennel (*n* = 67) were mostly professional breeders (52.2%), with only 25% of respondents having family-owned kennels. Lodging animals in the house was also significantly correlated to business size (*p* < 0.0001), since breeders raising less than ten bitches counted for 95.4% of the responses (and those raising up to five bitches for 84.6%).

Once the bitches approached whelping time, in almost half of the cases (46.7%), they were transferred to a maternity area. In France, by law (order of 3 April 2014), all dog breeders are required to provide a specific separated area for females one or two weeks before whelping and for the first weeks of postpartum (Figure 4). Females and their litters need to have access to a comfortable place, and not be in direct contact with the floor. The area should allow the dam to move away from her litter.

### 3.3. Common Practices Used around Breeding and Whelping Time

Regarding breeding and whelping management, surprisingly, over 50% of the respondents use progesterone levels measurement during estrous to determine the time of ovulation, while 12.7% used progesterone levels as well as vaginal smears. More than 60% of the surveyed breeders used progesterone levels as a breeding management tool (Figure 5).

On the other hand, to estimate the time of parturition, calculation was done by gathering information with no outstanding single method (Figure 6). Behavior changes most indicative of whelping preparation were refusal to eat/loss of appetite, restlessness, circling, scratching at the whelping box, looking for human contact, nest preparation, panting, and isolation.

## 4. Management, Signs of Stress and Behavior of the Bitch in the Peripartum Period

Labor and whelping usually occur under close observation of the breeder or another person. Less than 10% of the respondents relied only on electronic monitoring (video or acoustic) during parturition (Figure 7).

To characterize the level of stress experienced by the bitch at this time, the most common behaviors quoted by breeders were restlessness and/or excessive wandering (Figure 8). Other observed behaviors, such as trembling and panting normally associated with the physiology of parturition, were also described.

On the other hand, a non-overstressed bitch during whelping time, according to their breeders, will express it in three major and different ways (Figure 9): “searching for human contact” and in contrast “isolating and searching for a quiet place”. Sitting comfortably in the whelping box was also perceived as an indicator of a calm animal.

Because companion dogs are emotionally connected to humans, signs of distress are often minimized by the presence of the caregiver. More than 80% of respondents remain with their bitches around parturition to provide comfort (Figure 10). The size of the breeding operation significantly affected (*p* = 0.0006) the method chosen to address stress. Human contact was an option more often selected in small facilities (80.5% of those breeders raising less than 10 bitches) compared to larger facilities (58.5%), due most likely to a limitation and cost in human resources. To minimize stress at whelping time, playing music (15.7%) or using natural (non-drug) products (18%) were options cited at a similar frequency. Natural solutions were essentially homeopathy (5.5%), Bach flowers (5.2%) and a dog appeasing pheromone (Adaptil^®^) (4.6%).

After parturition, the majority of respondents reported that bitches will immediately display appropriate maternal behavior. For our sample, breeders considered a high frequency of nursing (85.5%) and licking puppies (82.1%) as adequate maternal behavior (Figure 11).

Only 10% of bitches, based on breeder responses, need more than two days to adjust to the maternal role. In fact, refusal to remain with the puppies is the main indicator of maternal overstress leading to insufficient care of puppies, followed by the frequent displacements of the puppies (Figure 12). Interestingly, only a small number of responders declared that their bitches never experienced stress or displayed signs of stress at parturition (5.5%, *n* = 19/345).

Inappropriate maternal behavior is described as “likely” by 67.1% of respondents, especially for primiparous bitches (96.1%). In terms of abnormal maternal behavior, 18.8% (65/345) of the breeders sometimes observe bitches refusing to care for puppies, primarily by ignoring them. This was significantly more often observed in larger kennels raising more than 10 breeding bitches than smaller ones (35.8% versus 15.8%, *p* = 0.010). A difference was also found according to the type of housing, with more problems reported in facilities housing bitches with no access to the home when not in peripartum, compared to bitches living inside the house or with access to it (26.9% versus 15.2% respectively, *p* = 0.024). A small proportion of breeders, 10.7% (37/345), have observed extreme maladjusted maternal behaviors (i.e., cannibalism of puppies). Again, this was significantly more often observed in larger kennels than smaller ones (28.3% versus 7.5%, *p* = 0.0004).

For more than 75% of responders, the primary method for controlling maternal anxiety after whelping is to stay in frequent contact with the bitch (Figure 13). Twenty-one percent (21.1%) of owners use natural products (mostly Bach flowers, 8.4%, or pheromones, 5.2%) to minimize stress after parturition, either as the only method (6%) or, most frequently, with other alternatives. Similarly, 18.2% of breeders play music in the maternity area to relieve stress.

In relation to the time of contact between the dam and the puppies, for around 90% (310/345) of the responders, the bitch will stay in contact with puppies for at least six up to nine weeks.

## 5. Discussion

Very few published studies provide detailed information on current dog breeders and their operations and practices (described here in the breeder profile). The importance of better knowledge of breeders and practices for producing pet dogs has been underestimated as well as the impact of maternal stress on puppy development. In addition, the role of veterinarians in the dog production process is less evident than for breeding programs for other animals, such as cows, horses and pigs. Our goal, here, was to describe the general characteristics and the opinions of dog breeders in relation to the reproductive process and maternal behavior characterization and overstress management in the peripartum period.

The representativeness of our sample (*n* = 345) compared to the overall population of dog breeders has been assessed using different approaches. Compared to the number of breeders registered at the French Kennel Club (Société Centrale Canine) the number of responses represented only 0.35% (345/98121) of breeders. However, this response rate (less than 1%), was similar to recent survey-based studies [15,18,19] but less than the response rate published by Leroy et al. (2007) [16]. The primary reason for these response rate differences lies in the methods used to select and contact dog breeders. In Leroy et al. 2007 [16], questionnaires were distributed by the French Kennel Club, while the other studies, including ours, relied on contacting breeders via email. The latter distribution method is less likely to be effective at motivating survey participation. On the other hand, the online questionnaire may also have pinpointed breeders that are more engaged, with greater technical-based management, thus potentially leading to a biased view of the overall practices on the field. A small proportion of our sample completed our survey after being contacted through the CERCA-EnvA mailing list (11.9%) or through the participant list of a roadshow co-organized in 2017 by Ceva and Royal Canin (5.7%). For all survey-based studies where participants are self-selected, controlling bias is impossible. Most likely, for responders to an online survey that is not mandatory and offers no reward, participation is limited to individuals that are most interested in the subject matter surveyed. Our findings, therefore, may not reflect the overall practices of dog breeders across France, but it is representative of a considered part of the sector.

The features of our sampling and results were similar to a study of the Australian Kennel Club [15]. For example, data from the Australian responders aligned with our results in that over half of the breeders (58.8%) considered their activity to be a familial, small-sized business involving a single dog breed (76.6%) with less than five reproductive bitches that produced one litter or less per year (64%). Although our sample did not include all dog breeds, the top breeds listed by the French Kennel Club were represented, as well as all the dog groups.

In the present study, bitches were usually kept inside the house, except for parturition and the peripartum period. Housing bitches inside the home was found to be correlated with breeding activities qualified as “familial” as opposed to professional business and logically more often selected by breeders raising fewer bitches (less than ten). From the 203 responders that considered their activity as “familial”, 150 housed the bitches inside the house (73.8%). Therefore, the term “familial”, is debatable. By French law until recent years, a breeder was considered to be any person with one or more breeding females and producing at least two litters per year. Therefore, the production of only one litter was not considered a commercial activity. A new text was reinforced in 2016 stating that once a puppy is sold per year, the person is labeled as breeder and must accept all that this entails, including the obligations and rules of hygiene or skills needed to exercise the dog breeding activity [20]. However, the denomination “familial” might bare a concept of puppies born within the house with almost all the time in human presence, which could imply better socialized animals. The definition of professional could lead to a characterization of puppies raised with less contact with people. Since the classification of breeding facility was auto denominative, it is not possible to verify the exact status of each responder. Overall, the housing systems employed by breeders varied from dedicated rooms in the house or adjacent rooms with direct access to the entire house to kennels or chalets in the garden. In some cases, bitches were allowed to freely circulate during the day but were kenneled at nighttime. Interestingly, whatever the type of business activity (familial or professional), human contact was a key factor, with 75% or breeders indicating that bitches have the opportunity to be in constant contact with their caregivers (Figure 3).

For the timing and management of the breeding process, measurement of progesterone levels was used by over 60% of responders. Although more expensive than other analyses, progesterone level assays are an accurate way to determine the timing and range of the fertile period during estrus [21,22]. Measurement of progesterone levels has also been suggested for bitches that have experienced previous reproductive problems and/or when successful breeding is imperative [23]. To our knowledge, however, the current use of progesterone measurement by breeders in the field has not been formally investigated. Interestingly, only 4.3% of breeders in our sample reported use of behavior cues for determining whether the female was within the window of fertility.

Breeder determination of parturition time employed a combination of methods, with the date of last breeding (42.1%) and the ovulation day (37.1%) being the most frequently mentioned estimators. Although there are a variety of methods to predict parturition in dogs, the most common methods used by breeders were in accordance with the literature [24,25,26]. The best predictors of impending parturition are ovulation date as determined by progesterone levels and the sharp drop of progesterone at the end of gestation.

For the vast majority of breeding facilities, whelping occurs under close supervision (94.2%), with a small percentage (17.1%) using both human presence and a video monitoring system. The presence of a familiar person seems to be reassuring for the bitch around whelping time. Considering that parturition is a complex moment, if the stress level is high, the bitch may suffer from a negative experience that could perhaps affect her maternal behavior. Overall, responders agreed that parturition is stressful for bitches with only 5.5% (19/345) stating that they had never observed maternal stress during parturition. A common indicator of stress was excessive wandering and agitation. In contrast, no specific behavior was identified to describe the status of animals not considered to be overstressed. From the most frequently quoted signs (Figure 9), the preference for one over the other might be related to the level of interaction between the bitch and the breeder. Therefore, dogs that have a closer bond to their owner might be in pursuit of human contact (56.5%), while animals not so used to human contact might prefer to isolate and find a quiet place (46.4%).

To manage stress, human presence and contact with bitches was the primary method cited. However, there was a significant difference (*p* = 0.001) in stress management strategies depending on the size of the kennel. To minimize females’ stress, human contact was more often selected by smaller facilities than larger (80.5% versus 58.5%), whereas larger operations used video surveillance systems (37.7%) more often than smaller breeding kennels (16.8%). Although anxiety has been shown to affect physiological changes, heart rate and contractions during parturition [27,28], very little research on modulating maternal stress during the peripartum period has been conducted. For the breeders surveyed, common sense dictated increased human contact to reduce maternal anxiety at parturition. While this approach does require significant human availability and resources, it has been shown that human interaction is effective in reducing anxiety and thereby improves the welfare of dogs in shelters [29]. Other calming techniques described by participants included playing music in the maternity area and using non-pharmaceutical (natural) products such as homeopathy, Bach flowers and dog appeasing pheromone (Adaptil^®^) For all breeders, methods to alleviate stress at parturition were selected either systematically and used as a preventative (40.7%) or selected on a case-by-case basis when overstressed females were identified (59.3%).

Breeders evaluated the quality of “motherly style” based on the frequency of two major behaviors, namely, nursing and licking the newborns. In fact, recent studies have shown that mothers who scored high on maternal care were more often in contact and showed high levels of oral behavior toward their puppies [30,31,32]. Breeders usually observed the role of motherly behavior by the bitch soon after parturition. Signs of overstress and/or poor maternal care were associated with rejection and frequent displacement of puppies. Over 60% of the responders had observed inappropriate maternal behavior, particularly in primiparous females. Differences in terms of maternal behavior, whether appropriate or not, might be observed according to the dog breed.

As in the prenatal and parturition stages, the most common method used to control maternal stress during the postpartum period was to increase the human presence for 77% of the survey sample. Methods or products to address their stress after parturition were selected on a case-by-case basis by 73.3% of responders (253/345), and only 26.6% used a systematic approach for relieving or preventing stress. Considering that 40.7% of responders used systematic procedures to reduce stress at the time of whelping, it is likely that breeders consider that stress is less likely to affect breeding females once puppies are delivered.

## 6. Conclusions

This study provides a synopsis of the opinions, management and operational practices implemented by French dog breeders for producing puppies. Although the sample might represent a specific portion of the universe of puppy production in France, the gathering of information is a step towards improving our understanding of a rather complex activity. In addition, the responders of the survey might be part of a group of breeders more inclined to invest in selection and more engaged in the new aspects of the activity. A better knowledge of the peculiarities of dog breeders’ profiles and the diversity of their practices in assisting in the reproductive process and the management of stressful situations during the peripartum and maternal care of offspring is important to characterize the activity and might have a direct impact in improving animal welfare. Although the effect of maternal care on the development of puppies was not directly addressed, the responders were very united in their descriptions of the signs of good quality maternal skills. The importance of providing an environment fostering appropriate conditions for the motherly behavior of the bitch to be fulfilled was, as well, a common concern amongst the responders. Considering the lack of general information regarding the organization of breeding facilities, breeders’ perception and management of potential stress in their dogs at different stages of the puppy production, our results provide insights that will allow veterinarians to better interact with breeders. Comprehending dog breeders’ motivation and practices is a way to help kennel clubs to address the problems in puppy production and consequently improve the health of the dam and puppies.

## Figures and Tables

**Figure 1 animals-10-00159-f001:**
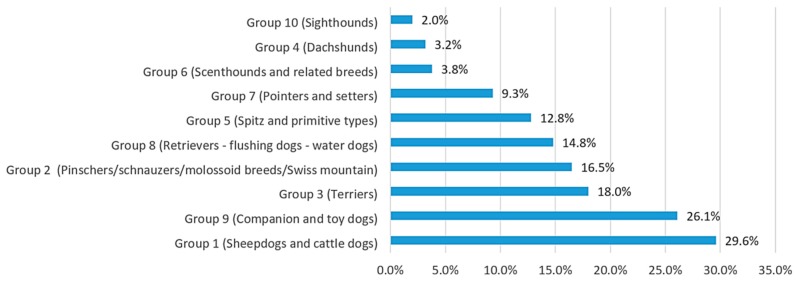
Representation of the breeds by dog group amongst the respondents (World Canine Organization).

**Figure 2 animals-10-00159-f002:**
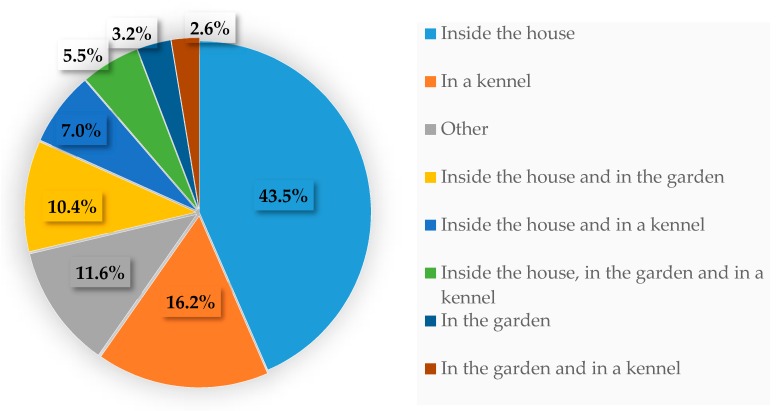
Housing systems of the breeding bitches when not in peripartum (multiple answers allowed).

**Figure 3 animals-10-00159-f003:**
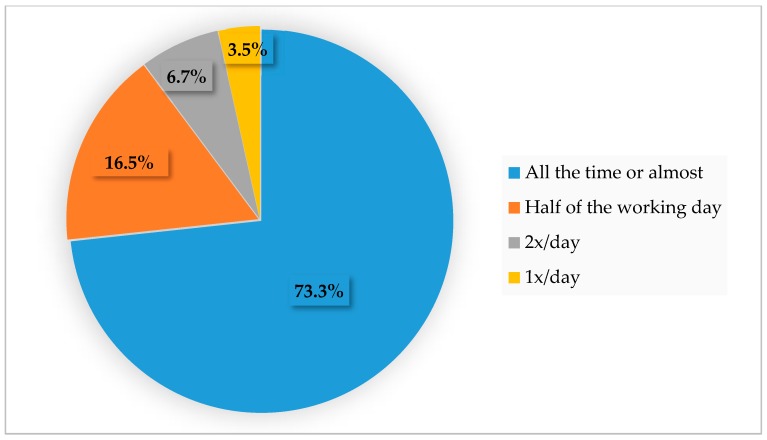
How often the breeding bitches were in contact with humans.

**Figure 4 animals-10-00159-f004:**
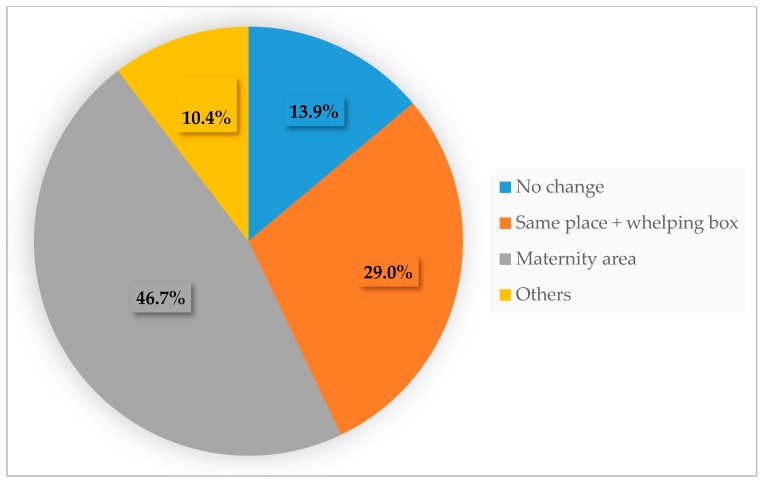
Housing of the breeding bitches during the peripartum period (percentages, multiple answers allowed).

**Figure 5 animals-10-00159-f005:**
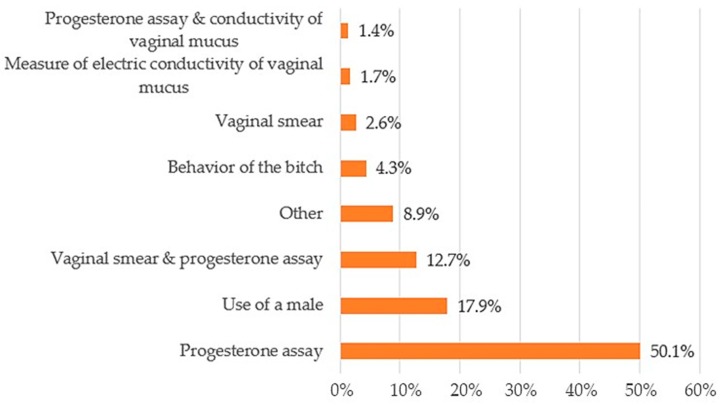
Techniques used to estimate the breeding time (multiple answers allowed).

**Figure 6 animals-10-00159-f006:**
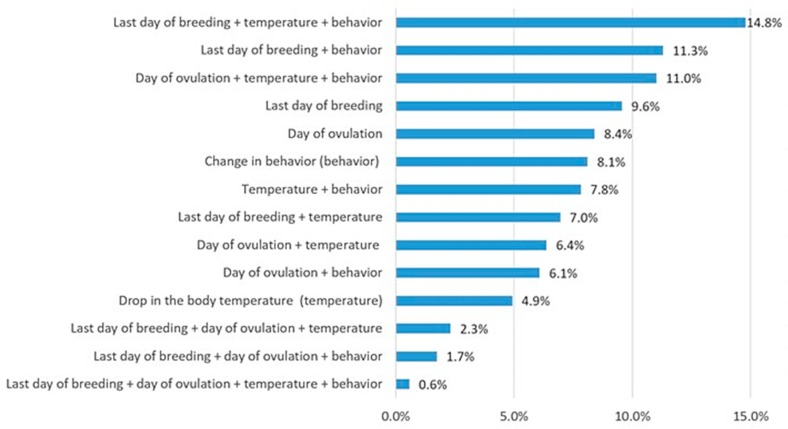
Methods used to estimate the whelping time of the bitch by using single or multiple information based on last breeding day, ovulation time, change in behavior and drop in the body temperature (multiple answers allowed).

**Figure 7 animals-10-00159-f007:**
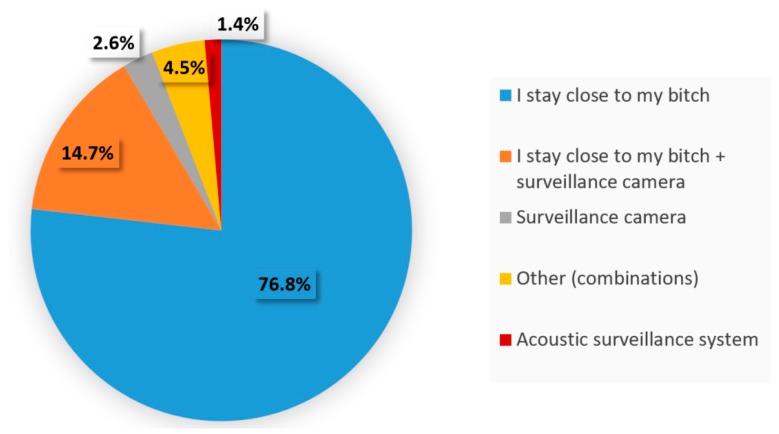
Methods of surveillance of the bitch at whelping time.

**Figure 8 animals-10-00159-f008:**
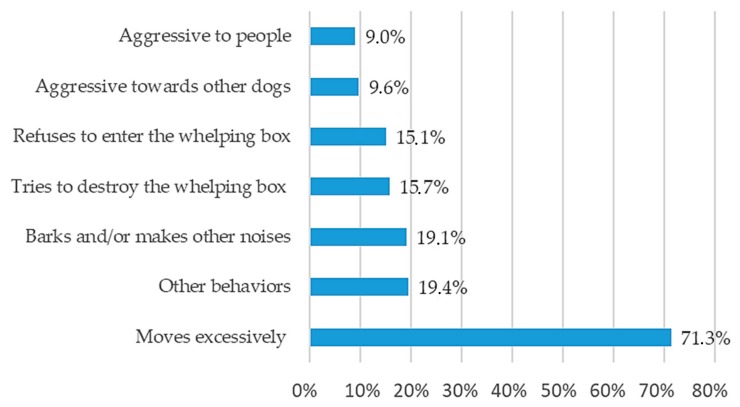
Common behaviors reflecting stress of the bitch at whelping in the breeders’ opinion (multiple answers allowed).

**Figure 9 animals-10-00159-f009:**
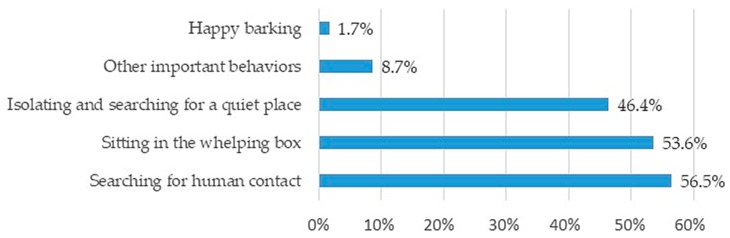
Signs of a non-stressed bitch during the whelping time (multiple answers allowed).

**Figure 10 animals-10-00159-f010:**
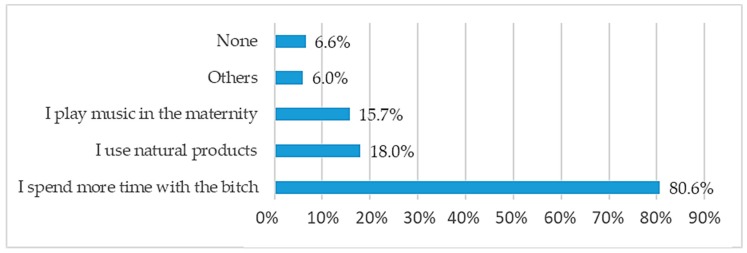
Ways to reduce stress in the bitch during parturition (multiple answers allowed).

**Figure 11 animals-10-00159-f011:**
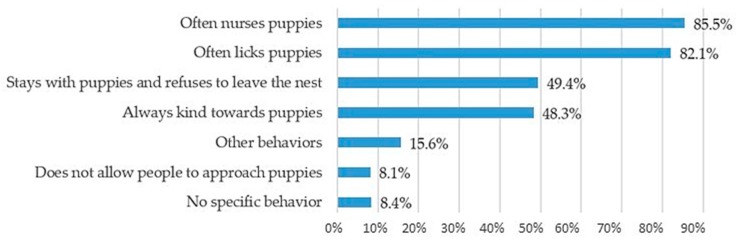
Signs of motherly attitude after puppies’ birth (multiple answers allowed).

**Figure 12 animals-10-00159-f012:**
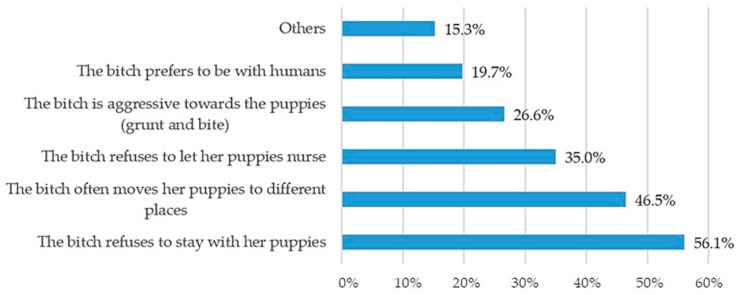
Common behaviors observed by breeders and perceived as signs of stress in bitches after parturition (multiple answers allowed).

**Figure 13 animals-10-00159-f013:**
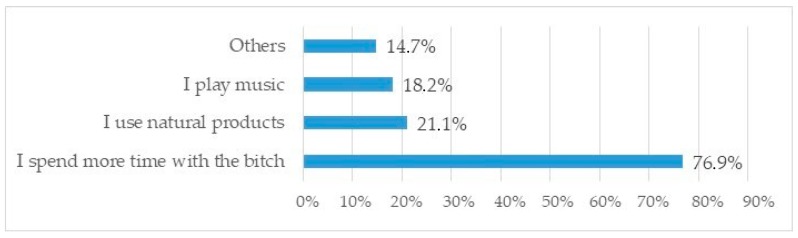
Methods to control anxious bitches after parturition (multiple answers allowed).

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
