# Peer review of "The View of the French Dog Breeders in Relation to Female Reproduction, Maternal Care and Stress during the Peripartum Period"

_animals, 2020, doi:10.3390/ani10010159_

Round 1

Reviewer 1 Report

No more comments 

Reviewer 2 Report

 I think the manuscript can be accepted in the present form. It has been well improved compared to the first submission. 

Kind Regards

This manuscript is a resubmission of an earlier submission. The following is a list of the peer review reports and author responses from that submission.

Round 1

Reviewer 1 Report

The topic of the paper "The view of the French dog breeders in relation to female reproduction, maternal care and stress during the peripartum period" is interesting and relevant considering the lack of information and data about dog  breeder practices in European countries.

I appreciate the work done by the authors even if I have some concerns about the sampling method used. Anyway, my position is mitigated by  the gap in published data about the field. This paper is not representative of the sector in France but it gives some clues useful to explore breeder perceptions and some procedures.

Introduction is quite nice. For what concerns the survey, I think it would be useful to have not only the number of litter/year but also the number of puppies/year, features of the maternity area, how long before whelping the bitch was transferred to the maternity area. 

I suggest the author to complete the section Ethics Statement referring to the GDPR.

I would like to improve analysis considering management differences linked to the dog breed (especially dog size and attitude) and breeder region (Northern or Southern France). It would be nice also to have respondent geographical distribution. I would also find interesting to analyse correlation between common behaviour reflecting stress in the bitch at whelping in the breeder's opinion  and type of management the breeder declared. I've not found the analysis of data collected in the open-ended section (about bahavioural differences among breeds).

Author Response

Dear Reviewer,

All the suggestions and corrections were addressed as indicated in the document (author-coverletter).

Thank you for your time and attention,

Reviewer 2 Report

This article wants to provide opinions, management and operational practices implemented by French dog breeders for producing puppies. The topic requires attention but some changes are necessary.

Line 49: Is the fourth position referred to the total dogs or the ratio?

Line 56: Is the SCC responsible just for puppies born in France?

Line 58: there is a space more in 27, 993

Line 138: results are reported in a no consistent way. Sometimes just percentages are reported, sometimes number of answer + percentage. Is there a reason for this?

Figures: are not reported in a consistent way. All figures look differents 

Numbers: decimals are reported in this way 00,00 or other way too 00.00. Please be consistent for this

Agressive: you reported several time the word agressive with one g. It should be "aggressive".

Line 186-187: this is not a results but I think it probably part of your discussion

Line 195: instead of owners, I think you referred to breeders

Line 196-197: is there a correaltion that proves this sentence "The difference in the behaviors seem to be associated with 196 the level of interaction between the bitch and the breeder."

Line 197: "attachment" requires a specific definition (see for example Topal et al, 1998). What do you mean in this case?

Line 199:  is the "sitting" and "being comfortable"the same?

Line 206: did you perform any statistic to prove that the frequency is the same?

Line 229: Is this part of the paper related to the parturition time or "after" parturition time?

Line 267-274: correlations can not be found in the results

Line 286: there is a space between 4.3 and %

Line 296: in the results 5.2% was 5.5% for "they had never observed maternal stress during parturition"

Line 301: be consisent in reporting results. Here there is new version 80.5 - n=291

LIne 300-304: I can not find this part in the results

Line 312-317: I can not find this part in the results

Line 323: there is a , after puppies

Author Response

(The authors gave the same response as above.)

Round 2

Reviewer 1 Report

I think the manuscript needs only some minor revisions to be ready for publication: editing, for example line 302 (fig instead of figure).

Reviewer 2 Report

The manuscript still needs some more change in order to be clear enough for publication. Please, be consistent!!

Definition of "small" and "large" facilities should be reported since you based some of the questions on that. Why did you use "10" as a threshold?

Please be consistent when you report numbers in figures and texts. Sometimes those appear with ",", sometimes with ".". And also for some terms like "behaviours" : look at figure 5 and 6

Numbers reported in figure 1 and 5 look different in size and font.

Line 174: is enough to report law in this way? do you need to give more details?

I think figures should be realized in the same format. See for example figure 4 and 7.

Line 221: "due to the human resources requirements (80.5% versus 58.5% respectively)". What do you mean with the percentage? are referred to the human resources requirements?

Line 226-228 : I think this sentence requires more attention: it is not clear if the comparison is related to the other facilities or to the period (maternity)

Line 248: what kind of correlation did you use here? this is not reported in the "statistical analysis" section.

Line 302: so far "figure" instead of "Fig" has been used

Line 327: I was not able to find any results related to "calm". Is the same as "minimize stress"?